# Non-Invasive Early Detection of Oral Cancers Using Fluorescence Visualization with Optical Instruments

**DOI:** 10.3390/cancers12102771

**Published:** 2020-09-27

**Authors:** Takamichi Morikawa, Takahiko Shibahara, Takeshi Nomura, Akira Katakura, Masayuki Takano

**Affiliations:** 1Department of Oral and Maxillofacial Surgery, Tokyo Dental College, Tokyo 102-8159, Japan; sibahara@tdc.ac.jp (T.S.); takano@tdc.ac.jp (M.T.); 2Oral Cancer Center, Tokyo Dental College, Chiba 272-8513; Japan; tanomura@tdc.ac.jp (T.N.); katakura@tdc.ac.jp (A.K.); 3Department of Oral Oncology, Oral and Maxillofacial Surgery, Tokyo Dental College, Tokyo 102-8159, Japan; 4Department of Oral Pathobiological Science and Surgery, Tokyo Dental College, Tokyo 102-8159, Japan

**Keywords:** oral cancer, oral squamous cell carcinoma, fluorescence visualization loss, oral potentially malignant disorders, oral cancer screening, optical Instrument, medical artificial intelligence

## Abstract

**Simple Summary:**

Oral cancer has a high mortality rate. Then, oral cancer screening is needed for early detection and treatment. Fluorescence visualization is non-invasive, convenient, and in real-time, and examinations can be repeated. Our study aimed to show the usefulness of oral cancer screening with fluorescence visualization. A total of 502 patients were performed using fluorescence visualization that was analyzed using subjective and objective evaluation. Results of this study, subjective evaluation for detection oral cancer was high sensitivity and low specificity, while objective evaluation using imaging processing analysis was high sensitivity and high specificity. Therefore, oral cancer screening using fluorescence visualization is useful for the detection of oral cancer. The widespread use of this screening can reduce the mortality rate of oral cancer.

**Abstract:**

Background: Oral cancer screening is important for early detection and early treatment, which help improve survival rates. Biopsy is the gold standard for a definitive diagnosis but is invasive and painful, while fluorescence visualization is non-invasive, convenient, and real-time, and examinations can be repeated using optical instruments. The purpose of this study was to clarify the usefulness of fluorescence visualization in oral cancer screening. Methods: A total of 502 patients, who were examined using fluorescence visualization with optical instruments in our hospitals between 2014 and 2019, were enrolled in this study. The final diagnosis was performed by pathological examination. Fluorescence visualization was analyzed using subjective and objective evaluations. Results: Subjective evaluations for detecting oral cancer offered 96.8% sensitivity and 48.4% specificity. Regarding the objective evaluations, sensitivity and specificity were 43.7% and 84.6% for mean green value, 55.2% and 67.0% for median green value, 82.0% and 44.2% for coefficient of variation of value, 59.6% and 45.3% for skewness, and 85.1% and 75.8% for value ratio. For the sub-analysis of oral cancer, all factors on objective and subjective evaluation showed no significant difference. Conclusions: Fluorescence visualization with subjective and objective evaluation is useful for oral cancer screening.

## 1. Introduction

Oral cancer has a high mortality rate [1,2], particularly at advanced stages, such as stage III and IV [3]. Early detection and early treatment for oral cancer are important for control [1,2]. Around 95% of oral cancers are oral squamous cell carcinomas (OSCCs) in histopathology [3]. In the oral cavity, ocular inspection and palpation are easily performed. Therefore, The World Health Organization (WHO) has proposed oral cancer screening mainly based on ocular inspection and palpation as a conventional oral examination (COE) to be performed by general dental practitioners (GPs) [4]. Ocular inspection is especially important for oral cancer screening. However, since various oral mucosal diseases occur in the oral cavity, it is often difficult to discriminate between them. In particular, OSCC may develop from oral potentially malignant disorders (OPMDs), such as oral lichen planus (OLP), leukoplakia, erythroplakia, and chronic candida [5]. Early detection and management of oral epithelial dysplasia (OED) in OPMDs is an important preventative step against malignant transformation, such as OSCC [5,6]. It is important to detect these subtle changes at an early stage.

In ocular inspection and palpation, one major limitation is the difficulty of differentiating between benign and high-risk lesions (HRL) [7], as early-stage OSCC (T1-2N0M0) cancer in situ (CIS) [5,8], and OED, may not present typical features, and a wide variety of oral mucosal diseases can present in various ways in the oral cavity [9]. 

Biopsy is the gold standard for definitive diagnosis. The final diagnosis is performed by pathological examination. However, this process is difficult for GPs to perform and invasive. To improve the safety and effectiveness of oral cancer screening [10], the procedure should be non-invasive or mildly invasive, simple, and repeatable. Forms of oral cancer screening include cytology [11], vital staining [12], and fluorescence visualization (FV) [13] in addition to COE. 

FV has some advantages. FV is non-invasive, simple, convenient, real-time, and repeatable using optical instruments (OIs) [14]. FV uses blue light (400–460 nm) to illuminate flavin adenine dinucleotide (FAD), nicotinamide adenine dinucleotide (NADH), and collagen cross-links (CCL) [15,16,17]. The normal mucosa is visualized as apple-green autofluorescence through a selective filter with these coenzymes, and CCL. This is called fluorescence visualization retention (FVR). On the other hand, the abnormal tissues, such as HRL and inflammatory diseases, exhibit decreased autofluorescence and appear as dark brown. The appearance of this color is called FV loss (FVL) [18]. FVL is caused by the absorption of a specific wavelength of blue light due to the decreases in FAD and NADH, breakdown of CCL, and angiogenesis. However, the evaluation of FV has been visual and subjective. Therefore, definitive results have been lacking [13,14]. 

The purpose of this study was to clarify the utility of subjective and objective evaluations using FV for oral cancer screening. Furthermore, the study aimed to improve the accuracy rate by combining them. 

## 2. Results

### 2.1. Patients Characteristics

A total of 502 patients were enrolled during this study period. Of the study cohort 276 (55.0%) patients were men, and 226 (45.0%) were women. The tongue was the most commonly affected site (259 patients; 51.6%), followed by buccal mucosa for 124 patients (24.7%). The mean age was 62.3 years (Appendix A). 

Oral cancer was the most common condition, affecting 161 patients (32.1%, including 149 patients with early OSCC, and 12 with CIS), while 235 had OPMDs (46.8%, including 123 patients with OLP, 102 with leukoplakia, 3 with erythroplakia, and 7 with chronic candida), and 106 had other diseases (Others; 21.1% including 32 patients with stomatitis, 24 with a benign tumor, and 50 with normal mucosa).

### 2.2. Conditions

The characteristics of each condition are showed in Table 1. The percentage of man patients with oral cancers, OPMDs, and Others was 53.4%, 57.0%, and 52.8%. The mean age was 62.5, 62.1, and 59.9 years. The tongue was the most commonly affected site for each condition. On the control site, the subjective and objective evaluation showed no significant difference (Table 1).

#### 2.2.1. Oral Cancers

In the histopathological diagnosis results, early OSCC grades 1, 2, and 3 were seen in 98 (65.8%), 36 (24.2%), and 15 patients (10.0%). Stage I OSCC (T1N0M0) was observed in 77 (51.7%) and stage II (T2N0M0) in 72 patients (48.3%) according to the TNM Classification of Malignant Tumours 8th Edition by the Union for International Cancer Control (UICC). Visual classification [19] of oral cancers, superficial spread, exophytic, and the endophytic type was possible for 39 (24.2%), 68 (42.2%), and 54 patients (33.6%). 

A typical case of OSCC is shown in Figure 1. OSCC shows non-uniform FVL and unclear FVL borders. 

In the subjective evaluation, FVR was seen in only 5 patients (3.2%), while FVL was observed in 156 patients (96.8%). In the objective evaluation, the area was 215,408 pixels, mean G value (MeanV) was 60.9, median G value (MediV) was 58.4, the standard deviation of G value (SD) was 12.6, coefficient of variation of G value (CV) was 0.21, skewness was 0.87, kurtosis was 2.6, value ratio (VRatio) was 68.9%.

#### 2.2.2. OPMDs

Following histopathological diagnosis, for OLP, chronic inflammation was observed 121 (98.4%) patients and OED in 2 (1.6%; 1 low grade and 1 high grade). Regarding leukoplakia, hyperkeratosis, OED, and papillary hyperplasia were seen in 61 (59.8%), 31 (30.4%; 16 low grade and 15 high grade), and 10 patients (9.8%), respectively. As for erythroplakia, high-grade OED was seen in all patients. Chronic candida, *Candida albicans* was also detected in all patients. 

Typical cases of OLP and leukoplakia are shown in Figure 2 and Figure 3. OLP shows FV acceleration (FVA) in lace-like leukoderma and clear margin uniform FVL in erythema. Leukoplakia shows clear margin FVA in leukoderma. Most cases exhibited FVA and no FVL. Unclear FVL boundaries were observed for OED. 

In the subjective evaluation, the FVL rate was 61.7%. For the objective evaluation, the area was 160,045 pixels, MeanV was 72.1, MediV was 69.8, SD was 9.5, CV was 0.14, skewness was 0.28, kurtosis was 3.3, and VRatio was 93.8%.

#### 2.2.3. Others

Following histopathological diagnosis, catarrhal stomatitis and viral stomatitis were seen in 22 (68.7%) and 10 patients (31.3%) with stomatitis. Patients with stomatitis did not include those with symptomatic disease, such as pemphigus, pemphigoid, or Behçet’s disease. A benign tumor, fibroma and papilloma, lipoma, granular cell tumor, and solitary fibrous tumor were seen in 15 (62.5%), 5 (20.8%), 2 (8.3%), 1 (4.2%), and 1 (4.2%) of the patients. 

Typical cases of stomatitis and benign tumor are shown in Figure 4 and Figure 5. Stomatitis shows uniform FVL around the central ulcer. In the surface plot, a clear boundary in the center and uniform FVA and a uniform FVL in the periphery are observed. The benign tumor showed FVR alone and no FVL.

In the subjective evaluation, FVL rate was 32.1%. In the objective evaluation, the area was 129,048 pixels, MeanV was 68.7, MediV was 68.0, SD was 10.1, CV was 0.15, skewness was 0.36, kurtosis was 6.3, and VRatio was 95.8%.

### 2.3. Analysis

#### 2.3.1. Comparison between Each Group

In the subjective evaluation, the highest rate was found for oral cancers (*p* = 1.6 × 10^−10^). 

Results of objective evaluation showed that oral cancers were the conditions largest in area (*p* = 9.9 × 10^−9^); lowest in MeanV and MediV (*p* = 0.005 and 0.007); and highest in SD and CV (*p* = 9.0 × 10^−9^ and 5.2 × 10^−12^). Skewness and kurtosis were not significant (*p* = 0.107 and 0.082). Oral cancers had the lowest VRatio (*p* = 2.6 × 10^−17^). 

#### 2.3.2. Detection of Oral Cancers

In subjective evaluation, sensitivity and specificity were 96.8% and 48.4% (*p* = 1.4 × 10^−23^). 

Table 2 shows the objective evaluation for detection of oral cancers by receiver operating characteristic (ROC) curve analysis. The area under the curve (AUC) was 0.578 in area, 0.651 in MeanV, 0.633 in MediV, 0.512 in SD, 0.820 in CV, 0.651 in skewness, 0.513 in kurtosis, and 0.827 in VRatio (Appendix A). MeanV and MediV, CV, skewness, and VRatio were significant factors, and cut-off was set at 52.5 in MeanV, 60.0 in MediV, 0.15 in CV, 0.71 in skewness, and 87.5% in VRatio. Sensitivity and specificity were 43.7% and 84.6% in MeanV, 55.2% and 67.0% in MediV, 82.1% and 69.4% in CV, 59.6% and 63.0% in skewness, and 85.1% and 75.8% in VRatio (Table 2).

As the result of multivariate analysis, subjective evaluation and CV, VRatio was a significant factor; Subjective evaluation, *p* = 0.004, odds ratio (OR) = 13.866, and 95% confidence interval (CI) = 3.098, 62.084; CV, *p* = 0.005, OR = 5.454, 95% CI = 2.871, 10.360; VRatio, *p* = 0.001, OR = 5.642, 95% CI = 2.928, 10.871 (Table 3).

#### 2.3.3. Combination of Each Factor for Detection of Oral Cancer

Each item was combined for subjective evaluation, and CV and VRatio were evaluated. Each OR was scored by the following calculation formula.

Combination for detection of oral cancer = 13.9 × Subjective evaluation + 5.5 × CV + 5.6 × VRatio

(minimum score 0, maximum score 25)

Using ROC curve analysis, the combination for the detection of oral cancer was set; the AUC was 0.891, and the cut-off was set at 19.4. Sensitivity and specificity were 86.6% and 84.6% (Appendix A). 

#### 2.3.4. Detection of OPMDs 

In the subjective evaluation, sensitivity and specificity were 58.5% and 26.4% for detection of OPMDs (*p* = 3.3 × 10^−4^). 

The objective evaluation for detection of OPMDs by ROC curve analysis is shown Table 4. The AUC was 0.450 in area, 0.602 in MeanV, 0.601 in MediV, 0.632 in SD, 0.740 in CV, 0.632 in skewness, 0.523 in kurtosis, and 0.767 in VRatio (Appendix A). MeanV and MediV, SD, CV, skewness, and VRatio were significant factors. Cut-off was set at 80.0 in MeanV, 50.0 in MediV, 12.9 in SD, 0.17 in CV, 0.56 in skewness, and 74.5% in VRatio. Sensitivity and specificity were 39.9% and 74.8% in MeanV, 81.3% and 36.8% in MediV, 83.5% and 38.1% in SD, 80.2% and 56.7% in CV, 68.8% and 64.8% in skewness, and 82.7% and 55.7% in VRatio. 

As seen the results of multivariate analysis, subjective evaluation, CV, and VRatio were significant factors (subjective evaluation, *p* = 1.6 × 10^−4^, OR = 0.089, and 95% CI = 0.025, 0.314; CV, *p* = 0.012, OR = 0.001, 95% CI = 0.000001, 0.188; VRatio, *p* = 0.031, OR = 1.024, 95% CI = 1.002, 1.046; Table 5).

## 3. Discussion

Early diagnosis by GPs is considered likely to improve the outcomes of oral cancer. A biopsy is the gold standard method of definitive diagnosis for oral cancer. However, the procedure is highly invasive, and painful. GPs may require specialized training to perform a biopsy, and the dissemination of cancer cells into the circulation results in an increased risk of metastasis after the biopsy [20,21]. Screening should be minimally invasive, inexpensive, and repeatable. Cytology [11], vital staining [12], and FV [13,14,15,16,17,18,19] are simple and effective methods of screening for oral cancer. 

Oral liquid-based brush cytology (OLBC) is an approach that involves collecting cells from the oral mucosa using a brush technique and creating homogeneous slides, and this method in efficient in cell collection [22]. OLBC is easy to use, relatively painless, and well-accepted by patients [23]. It was reported that 69.0% and 29.0% of patients feel discomfort and pain, respectively, with OLBC [24]. Several days are required to obtain results from OLBC. OLBC is also not a reliable means of evaluating lesions with thick keratin layers [22]. According to reports [11,22,23,24,25,26], the sensitivity and specificity are 75.0% and 50.0% [22], and 86.5% and 94.3% [26]. 

Vital staining is a method that can detect suspicious lesions using iodine solution (IS) and toluidine blue (TB) [12]. Vital staining with IS and TB in the oral cavity allows easy observation of the results in real-time. In OSCC, CIS, and low- and high-grade OED, little glycogen is present in the granule cell layer, resulting in iodine unstained (IU) areas [12,27]. However, IS cannot be used for patients with iodine allergy, and the technique is invasive [28]. IS is applicable for movable mucosae such as the tongue, oral floor, buccal mucosa, and soft palate, but is difficult to use on keratinized mucosa such as the gingiva and hard palate [10]. It was reported that 53.1% and 21.4% of patients feel discomfort and pain, respectively, with IS [28]. According to reports [12,27,28,29], the sensitivity and specificity for detection oral cancer are 56.0% and 46.8% [29], and 88.5 and 51.6% [12]. TB binds to the acidic groups of acidic mucopolysaccharides, such as nucleic acids of DNA and RNA, and exhibits metachromasia, dark royal blue relative to OSCC and CIS, a pale blue on high-grade OED, and unstained on normal mucosa, depending on nucleic acid content [12,30]. However, TB has invasive aspects such as sourness and bitterness and an acetic acid odor. TB is hazardous if swallowed and was shown to have toxicity to fibroblasts [31]. Inflammatory lesions bind TB, and this can contribute to false-positive outcomes. TB is applicable for all oral mucosa in detecting only oral cancer, but it is debatable whether OED can be detected with this method [32]. TB is also not reliable for evaluating lesions with thick keratin layers [30]. According to reports [12,33,34], the sensitivity and specificity for detection oral cancer are 80.8% and 96.7% [12], and 100% and 36,9% [35]. 

FV with OIs is simple and non-invasive, convenient, and examinations can be repeated [13,14]. The results of FV are available in real-time. It can be adapted to any area of the oral cavity (Figure 6). There are various types of OIs, such as VELscope^®^ (LED Medical Diagnostics, White Rock, BC, Canada) [35], ViziLite^®^ (Zila Pharmaceuticals, Phoenix, AZ, United States), ORALOOK^®^ (HITS PLAN, Tokyo, Japan), and IllumiScan^®^ (SHOFU, Kyoto, Japan) [13]. ORALOOK^®^ is a lightweight and simple instrument; it has a built-in filter, while FV images and oral conditions are captured using one unit [7]. This OI irradiates the target with a blue light at an excitation wavelength of about 422 nm, while it can detect only apple-green fluorescence via a filter (520–560 nm). IllumiScan^®^ is a lightweight and simple instrument; it has a built-in filter, and only FV images are captured using one unit [16]. This OI’s excitation wavelength is 425 nm, its filter is 470–580 nm, and it can detect only apple-green fluorescence. These are the OIs that we developed jointly with our departments.

FV can enhance the visibility of oral mucosal abnormalities by activating tissue autofluorescence. Autofluorescence is due to the presence of endogenous fluorophores in cells, such as FAD, NAD, and CCL, which produce a fluorescent emission when exposed to the light of a specific wavelength [36]. The denaturation and destruction of CCL by dysplastic progression cause FVL [37]. The fluorescence intensity due to FAD decreases with the progression of dysplasia [14]. FAD and NADH are important fluorophores that become excited at those wavelength intervals in epithelium cells. These coenzymes are known to involve different types of intracellular energy metabolism, such as glycolysis, the tricarboxylic acid cycle, and the electron transport chain [38,39]. It is known that dysplastic progression enhances a form of anaerobic metabolism called the Warburg effect [40]. Since FAD and NADH are intermediate enzymes, they are consumed when anaerobic metabolism is enhanced, and, as a result, the autofluorescence decreases. FVL occurs because the wavelength of blue light is absorbed by hemoglobin. Furthermore, angiogenesis occurs due to cell proliferation [41], and an inflammatory response is triggered by the immune response, resulting in FVL [42].

Various accuracy levels have been reported; the sensitivity and specificity for detection oral cancer were 33.3% and 88.6% [42], 100% and 80.8% [43], and 91.0% and 100% [44]. The result of subjective evaluation in this study showed high sensitivity (96.8%) and low specificity (48.4%). As this difference in accuracy was for subjective evaluation, it was considered that there is a large difference between evaluators. Therefore, in this study, we performed subjective and objective evaluation by using image processing analysis. 

RGB is a color representation method. Various colors can be represented by the three primary colors of Red, Green, and Blue [45]. When the dominant wavelength of the light is in the upper end of the visible spectrum, it is perceived as red, and if it is in the lower end, it is defined as blue [46]. The human eye is more sensitive to green because the ratio of luminosity function is 555 nm for humans [47]. Hence, the OIs used in this study were selected because they use a single green fluorescence color (G value), and image processing analysis is simple. 

As a result of objective evaluation in this study, we set various factors, such as MeanV and MediV. MeanV was reported to contribute to the detection of oral cancer [15,16,48]. On the other hand, Kozakai also reported that it is difficult to distinguish oral cancer from OLP [49]. In this study, MeanV (60.9) and MediV (58.4) for oral cancers were lower than for OPMDs (MeanV 72.1, MediV 69.8) and Others (MeanV 68.7, MediV 68.0). The AUC of MeanV and MediV was 0.651 and 0.633. The sensitivity and specificity were 43.7% and 84.7% for MeanV, 55.2% and 67.0% for MediV. The results showed high specificity and low sensitivity. 

The variation was evaluated by SD and CV. SD is difficult to compare if the means are different. On other hand, CV can relatively evaluate the relationship of variation. Oral cancer is a highly variable disease due to the heterogeneity within the tumor [50,51,52]. Kosugi et al. reported that the G value had more variation and a wider range with increasing tumor progression using an animal tongue carcinogenesis model [52]. In addition, they considered that tumor progression can be monitored by this new G value analysis method in humans. In this study, CV in oral cancers (0.21) was high compared with other diseases (OPMDs 0.14, Others 0.15). The AUC of CV was 0.820. The sensitivity and specificity were 82.1% and 69.4% for CV. The results showed high sensitivity and high specificity.

The skewness and kurtosis show the distribution. There have been no reports on skewness and kurtosis in FV image processing analysis. In oral cancers, the skewness and kurtosis varied greatly depending on the case. Skewness for oral cancers was high compared with that for OPMDs and Others. Kurtosis for oral cancers was low compared with that for OPMDs and Others. The AUC of skewness and kurtosis were 0.651 and 0.513. The sensitivity and specificity were 59.6% and 45.3% for skewness.

FV is greatly affected by environmental factors such as natural light [45,53]. In this study, we tried to maintain the same conditions as much as possible. Therefore, the cases were in a state where there was no significant difference (Table 1, control). However, it is difficult to eliminate environmental factors completely in clinical practice. Therefore, in this study, we corrected it by using a normal mucosa as a control, represented by VRatio [7,15,16,17,49]. In this study, VRatio for oral cancers (68.9%) lower than that of OPMDs and Others (OPMDs 93.8%, Others 95.8%). The AUC of the VRatio was 0.827. The sensitivity and specificity were 85.1% and 75.8% on VRatio. There were high sensitivity and high specificity. The results of multivariate analysis for the detection of oral cancer showed that subjective evaluation, CV, and VRatio were significant factors. These combinations were effective for the detection of oral cancer.

In this study, OIs were used randomly. These OIs can be used equally in both subjective and objective evaluations [54]. It was considered that this semi-quantification would enable more accurate diagnosis assistance without being influenced by the evaluator. This study may be able to aid remote diagnostics, especially in under-developed areas and developing countries. Many reports have described the potential application of artificial intelligence (AI) to medical fields in recent years [55]. Using this semi-quantitative evaluation, development into Medical AI is expected [54]. FV can be used in any subsite [56]. 

There are several limitations to our study. First, this study used a small number of patients and was retrospective. Second, a detailed examination of OED was not conducted. OED is at risk of malignant transformation and is HRL. It is also necessary to consider the detection of HRL [7]. Therefore, we are planning prospective studies for the detection of HRL and treatments for other oral subsites.

FV is a non-invasive method for early detection of oral cancer. It is effective for oral cancer screening in addition to COE, vital staining, and OBC.

## 4. Materials and Methods 

A total of 502 patients were enrolled at Tokyo Dental College Chiba Hospital and Chiba Dental Center from January 2014 to December 2019. The final diagnosis was performed by pathological examination. The inclusion criteria were: (1) COE and FV were performed before treatment, (2) informed consent to participate in this study was provided, and (3) confirmed diagnosis was obtained by biopsy, except for stomatitis and normal mucosa. All patients with OSCC were re-staged using the 8th edition of the TNM staging system by the UICC [5,8]. The study was approved by the Ethics Review Committee of Tokyo Dental College (740) and performed in accordance with the requirements of the Declaration of Helsinki (64th WMA General Assembly, Fortaleza, Brazil, October 2013).

OIs randomly used in this study were ORALOOK^®^ and IllumiScan^®^. This was left to the attending physician. These OIs allowed FV images to be obtained and saved to the camera as digital data. FV images of the lesion can be observed in real-time because the OIs have their monitor. FV images were taken a room that was as dark as possible. The irradiating light was set perpendicular to the lesion and the irradiation range was set to about 10 cm [15,16,17].

In subjective evaluations, oral conditions and FV images were compared to evaluate FVR, FVA or FVL in lesions (Figure 7A,B) [57,58]. Image processing analysis was performed on the FV images using Image J software version 1.5 (National Institutes of Health, Bethesda, Maryland, USA) [59,60]. Regions of interest (ROIs) for the lesion and control area were established in FV images [15,16,17]. 

The ROI of the lesion was defined as the area of FVL. The control ROI was set at a distance away in the same sub-site of the oral cavity in normal mucosa in front of the lesion without FVL and oral mucosal lesions (Figure 7C). The state of the lesion ROI was expressed in the surface plot, and color mapping was performed for the lesion ROI (Figure 7D). In subjective evaluation, we defined a positive case as FVL and a negative case as only FVA or FVR by two or more screenings using the surface plot as a reference [15,16,17]. The FVL rate was calculated. Sensitivity and specificity were calculated as follows; [sensitivity = (FVL positive cases of oral cancers/all cases of oral cancers) × 100%], [specificity = (FVL negative cases without oral cancers/all cases without oral cancers) × 100%]. 

For differential diagnosis of oral mucosal diseases, the chi-squared test, Mann–Whitney U test, and Fisher’s exact test were used for statistical analyses. Cut-off values were set using an ROC curve to detect oral cancers and OPMDs. AUC, sensitivity, and specificity were calculated by ROC analysis. All statistical analyses were performed using SPSS version 27.0 (IBM, Tokyo, Japan). Values of *p* < 0.05 were deemed statistically significant. For the multivariate analysis, the factors that were significant in the univariate analysis were used. OR and 95% CI were calculated. During the combination of each factor for the detection of oral cancers, significant factors were extracted from the results of multivariate analysis. Each factor was weighted with reference to the OR, and the effectiveness of the combination was verified.

## 5. Conclusions

FV is a non-invasive method for the early detection of oral cancers. Subjective analysis showed high sensitivity and low specificity, and objective analysis resulted in high sensitivity and high specificity for the detection of oral cancers. 

## Figures and Tables

**Figure 1 cancers-12-02771-f001:**
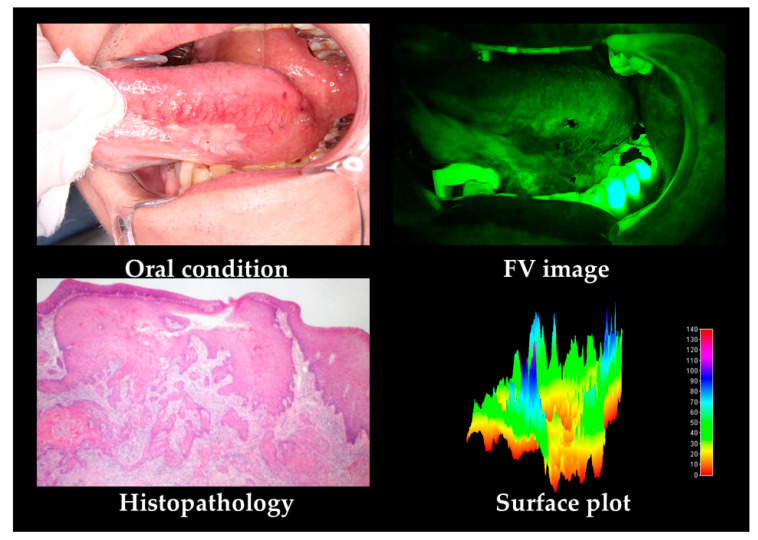
Grade 1 oral squamous cell carcinomas (OSCC) case according to histopathological features. As OSCC shows non-uniform and unclear margin fluorescence visualization loss (FVL). In the surface plot, non-uniform and unclear borders of decreased value are observed.

**Figure 2 cancers-12-02771-f002:**
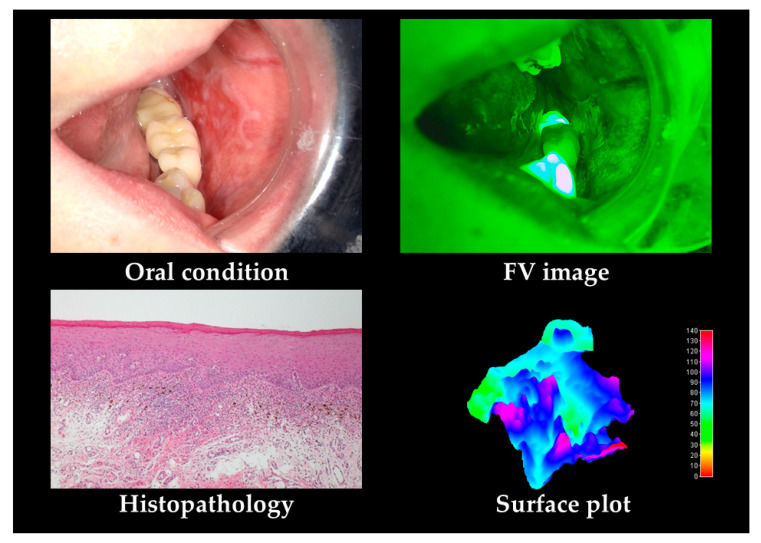
Oral lichen planus (OLP) case. Histopathological features include epithelial parakeratosis and zonal infiltration of subepithelial inflammatory cells. OLP shows fluorescent visualization acceleration (FVA) in lace-like leukoderma and clear margin, uniform FVL in erythema. In the surface plot, part of the erythema shows a decrease, and part of the lace-like leukoderma shows an increase.

**Figure 3 cancers-12-02771-f003:**
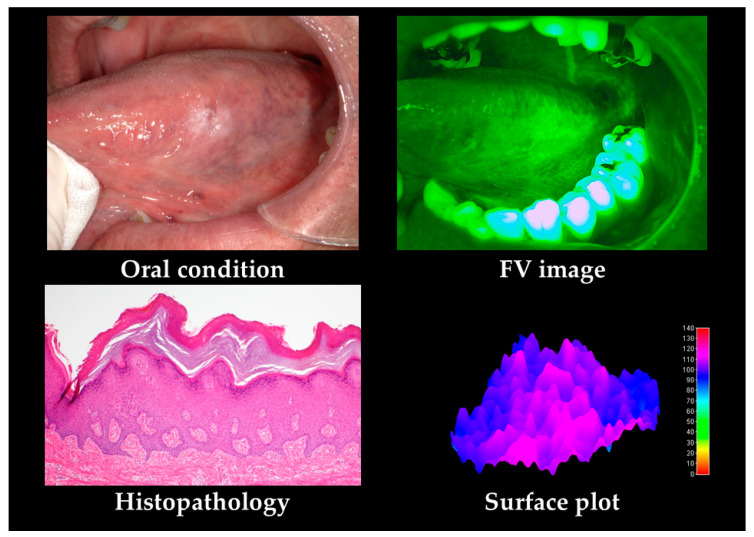
Leukoplakia case. The main histopathological feature is hyperplasia. The figure shows FVA alone and no FVL. In the surface plot, a clear boundary and uneven acceleration are observed.

**Figure 4 cancers-12-02771-f004:**
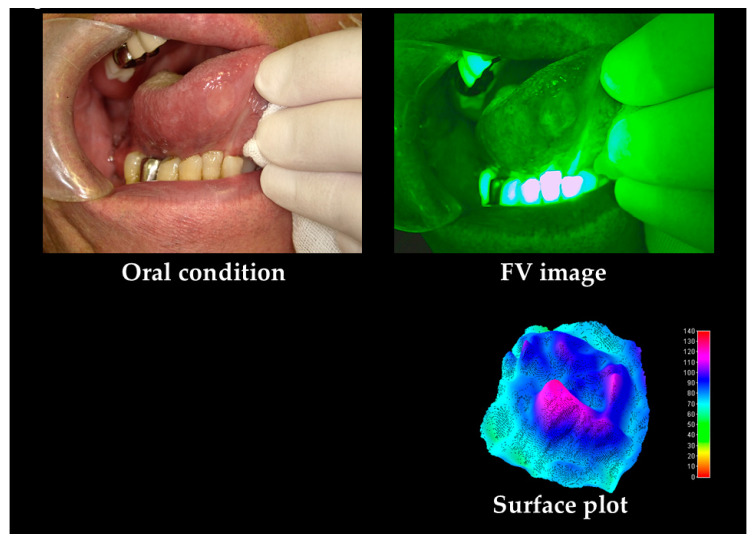
Stomatitis case. Stomatitis shows concentric narrow, clear margin, and uniform FVL around the central ulcer. In the surface plot, a clear boundary in the center and uniform FVA and a uniform FVL in the periphery are observed.

**Figure 5 cancers-12-02771-f005:**
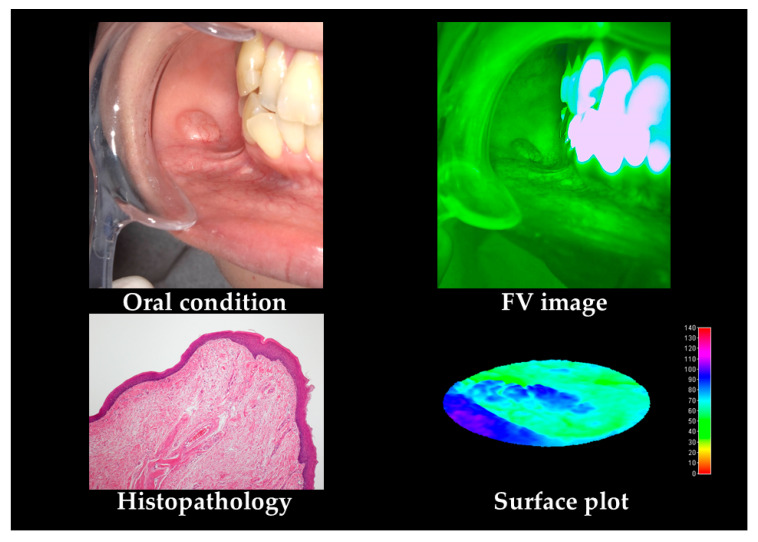
Fibroma as a benign tumor. Fibroma shows FVR alone and no FVL. The surface plot was observed to be flat.

**Figure 6 cancers-12-02771-f006:**
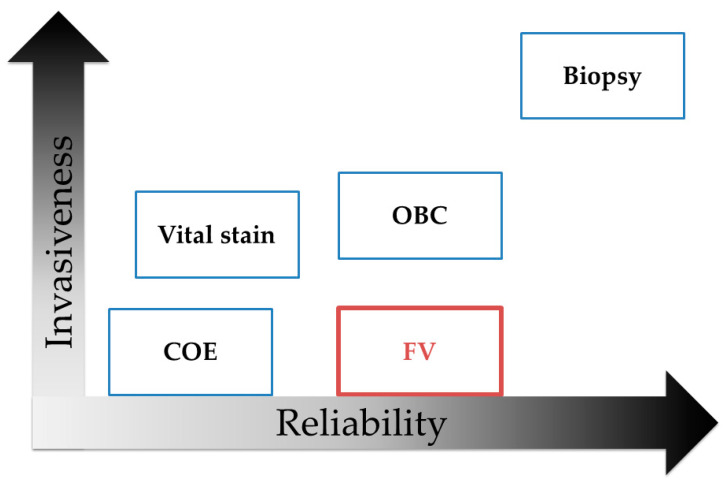
The concept of oral cancer screening in terms of invasiveness and reliability. Biopsy is the gold standard method for oral cancer screening. Conventional oral examination (COE) and FV are non-invasive methods, and FV is of high reliability.

**Figure 7 cancers-12-02771-f007:**
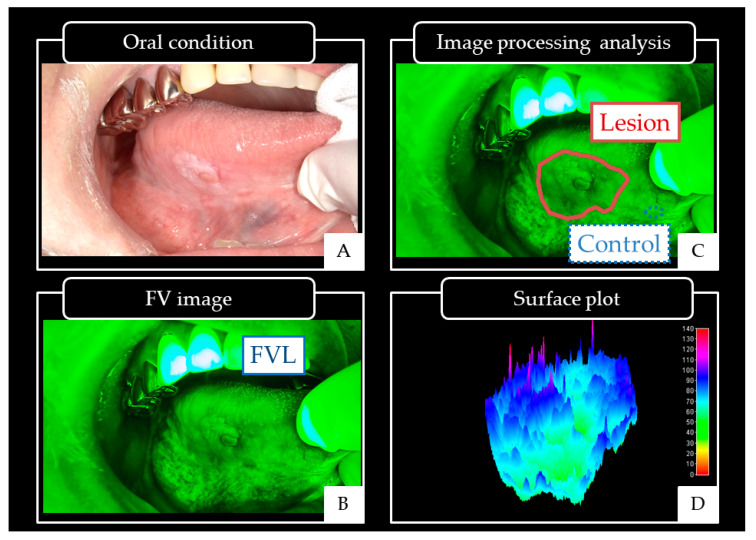
Oral conditions and FV image of cancer on the right side of the tongue.

**Table 1 cancers-12-02771-t001:** Patients’ characteristics on each condition.

	Oral Cancers	OPMDs	Others	*p*-Value
	*n* = 161	*n* = 235	*n* = 106	
Sex, men/women	86/75	134/101	56/50	
Age, mean	62.5	62.1	59.9	
Site Tongue Buccal mucosa Gingiva Plate Others	109 20 22 7 3	101 89 37 6 2	49 15 14 20 8	
Control site, mean FVL rate, (%) Area, (pixels) MeanV MediV SD CV Skewness Kurtosis	0% 927 81.9 80.2 2.9 0.05 16.8 −599	0% 967 81.7 80.2 2.9 0.04 78.1 −621	0% 962 80.9 80.1 2.8 0.04 55.9 −689	1.000 0.804 0.224 0.896 0.383 0.700 0.780 0.829
Lesion site, mean FVL rate, (%) Area, (pixels) MeanV MediV SD CV Skewness Kurtosis VRatio, (%)	96.9% 215,408 60.9 58.4 12.6 0.21 0.87 2.6 68.9%	61.7% 160,045 72.1 69.8 9.5 0.14 0.28 3.3 93.8%	32.1% 129,048 68.7 68.0 10.1 0.15 0.36 6.3 95.8%	1.6 × 10^−10^ 9.9 × 10^−9^ 0.005 0.007 9.0 × 10^−9^ 5.2 × 10^−12^ 0.107 0.082 2.6 × 10^−17^

MeanV: mean G value, MediV: median G value, SD: standard deviation, CV: coefficient of variation, VRatio: value ratio, OPMDs: oral potentially malignant disorders.

**Table 2 cancers-12-02771-t002:** Object evaluation from receiver operating characteristic (ROC) analysis for detection oral cancers.

	AUC	*p*-Value	Cut-off	Sensitivity	Specificity	95% CI
Area	0.578	0.066	-	-	-	0.494, 0.663
MeanV	0.651	5.0 × 10^−4^	52.5	43.7	84.6	0.570, 0.732
MediV	0.633	0.002	60.0	55.2	67.0	0.551, 0.715
SD	0.512	0.779	-	-	-	0.427, 0.597
CV	0.820	8.4 × 10^−5^	0.15	82.1	69.4	0.782, 0.858
Skewness	0.651	4.2 × 10^−4^	0.71	59.6	63.0	0.589, 0.747
Kurtosis	0.513	0.707	-	-	-	0.570, 0.731
VRatio	0.827	5.3 × 10^−14^	87.5	85.1	75.8	0.763, 0.890

ROC: Receiver operating characteristic, AUC: area under the curve, CI: confidence interval

**Table 3 cancers-12-02771-t003:** Multivariate analysis for detection oral cancers.

	Univariate		Multivariate	
	*p*-Value	*p*-Value	OR	95% CI
Subjective evaluation	1.4 × 10^−23^	0.004	13.866	3.098, 62.084
MeanV	5.0 × 10^−4^	0.329	1.066	0.814, 1.071
MediV	0.002	0.321	1.073	0.933, 1.233
CV	8.4 × 10^−5^	0.005	5.454	2.871, 10.360
Skewness	4.2 × 10^−4^	0.357	1.238	0.788, 1.938
VRatio	5.3 × 10^−14^	0.001	5.642	2.928, 10.871

OR: odds ratio.

**Table 4 cancers-12-02771-t004:** Objective evaluation from ROC analysis for the detection oral potentially malignant disorders (OPMDs).

	AUC	*p*-Value	Cut-off	Sensitivity	Specificity	95% CI
Area	0.450	0.063	-	-	-	0.397, 0.502
MeanV	0.602	1.7 × 10^−5^	80.0	39.9	74.8	0.551, 0.653
MediV	0.601	0.020	50.0	81.3	36.8	0.518, 0.684
SD	0.632	1.0 × 10^−6^	12.9	83.5	38.1	0.580, 0.683
CV	0.740	7.5 × 10^−19^	0.17	80.2	56.7	0.695, 0.786
Skewness	0.632	0.002	0.56	68.8	64.8	0.551, 0.713
Kurtosis	0.523	0.585	-	-	-	0.440, 0.607
VRatio	0.767	1.4 × 10^−10^	74.5	82.7	55.7	0.619, 0.712

**Table 5 cancers-12-02771-t005:** Multivariate analysis for detection OPMDs.

	Univariate		Multivariate	
	*p*-Value	*p*-Value	OR	95% CI
Subjective evaluation	3.3 × 10^−4^	1.6 × 10^−4^	0.089	0.025, 0.314
MeanV	1.7 × 10^−4^	0.249	1.078	0.949, 1.225
MediV	0.020	0.214	0.926	0.821, 1.045
SD	1.0 × 10^−6^	0.933	1.012	0.764, 1.341
CV	7.5 × 10^−19^	0.012	0.001	0.000001, 0.188
Skewness	0.002^4^	0.700	0.933	0.656, 1.327
VRatio	1.4 × 10^−10^	0.031	1.024	1.002, 1.046

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
