# Peer review of "Non-Invasive Early Detection of Oral Cancers Using Fluorescence Visualization with Optical Instruments"

_cancers, 2020, doi:10.3390/cancers12102771_

Round 1
Reviewer 1 Report
Major comments:
- The AUC in ROC analysis is too small (~ 0.65) for oral cancer. It should be above 0.7 or 0.8.
- The AUC in ROC analysis for OPMD is not found. The author may need to compare the AUC between OPMD and oral cancer.
- This method should be compared with the current method for Fluorescence Visualization.
Minor comments:
- Abstract: Please remove the ().
- line 361: 5. Conclusion is too short.
Author Response
Thank you for your advice.
Major comments:
- The AUC in ROC analysis is too small (~ 0.65) for oral cancer. It should be above 0.7 or 0.8.
→ In this study, the most significant VRatio was AUC 0.827 for oral cancer.
- The AUC in ROC analysis for OPMD is not found. The author may need to compare the AUC between OPMD and oral cancer.
→ We added the AUC in ROC analysis for OPMDs (Supplementary).
- This method should be compared with the current method for Fluorescence Visualization.
→ The accuracy of other fluorescent optical instruments has already been described in L346-348, 368-408.
Minor comments:
- Abstract: Please remove the ().
→ Thank you for your advice. Removed the () on abstract.
- line 361: 5. Conclusion is too short.
→ Thank you for your advice. Added on conclusion.
Reviewer 2 Report
Interesting paper and large series of patients, but the manuscript requires extensive editing and serious clarification of the Material and Methods and presentation of the results otherwise it cannot be suitable for publication.
-English editing is mandatory and will help reviewing the paper
-Abstract should be re-structured: the 1), 2) and 3) structure is unusual
-Material and Methods section: is the study referenced in clinicaltrials.gov? What was the primary endpoint of the study to conclude about the usefulness of fluorescence screening? The authors state that ORALOOK and IllumiScan were used randomly: what does it mean? Were patients randomized to one or the other method or was the method center dependent (in my understanding, two centers included patients)? What “subjective” (sometimes refered to “subject” evaluation in the text) and objective evaluation do mean exactly, it is extremely confusing to me.
-Page 2 lines 48-49: clinical evaluation with or without fluorescence is unable to assess the risk of transformation of an OPMD; this has to be clearly stated in the manuscript to avoid misleading the reader
-Page 2 lines 72-73 and table 1: contrasting “tongue” and “buccal mucosa” is not correct: the free border of tongue is included in the oral cavity, the base of tongue is not in the oral cavity
-Table 1 and 2 are redundant and I would keep Table 2 in the main text and move table 1 in supplementary
-Table 2: a relevant statistical test to evaluate the significance of the differences observed in the Control site and in the Lesion site needs to be added
-When an absolute number of patients is indicated throughout the text, a percentage should be added
-ROC curves should be provided as supplementary material
-It is unclear how the coefficients in the “combination of each factors for detection of oral cancer” (lines 160-166, p11) were defined
-All the paragraph “2.4.1. Sub-analysis of oral cancer” should be removed or added to supplementary material, as it does not bring much and is not the main message of the paper
-To many Figures are presented, they should be selected and the remaining moved to supplementary
Author Response
Thank you for your advice. Please give us a review.
-English editing is mandatory and will help reviewing the paper
→ We were re-edited English by MDPI.
-Abstract should be re-structured: the 1), 2) and 3) structure is unusual.
→ We corrected the re-structure.
-Material and Methods section: is the study referenced in clinicaltrials.gov?
→ Yes. But FV-related studies were limited to surgery in clinicaltrials.go.
There was No FV-related study for oral cancer screening, so this study were original and unique.
FV-related studies are limited to surgery, and screening is original in this study.
What was the primary endpoint of the study to conclude about the usefulness of fluorescence screening?
→ The primary endpoint of the study was to investigate the ability for screening oral cancer.
The authors state that ORALOOK and IllumiScan were used randomly: what does it mean? Were patients randomized to one or the other method or was the method center dependent (in my understanding, two centers included patients)?
→ We randomized patients using OIs. It was left to the attending physician.
What “subjective” (sometimes refered to “subject” evaluation in the text) and objective evaluation do mean exactly, it is extremely confusing to me.
→ The text has been unified to “Subjective evaluation”.
The subjective evaluation is the presence or absence of FVL. The objective evaluation is an evaluation of digitization by image processing analysis.
-Page 2 lines 48-49: clinical evaluation with or without fluorescence is unable to assess the risk of transformation of an OPMD; this has to be clearly stated in the manuscript to avoid misleading the reader
→ We added on L61-62 “It is important to detect these subtle changes at an early stage”.
-Page 2 lines 72-73 and table 1: contrasting “tongue” and “buccal mucosa” is not correct: the free border of tongue is included in the oral cavity, the base of tongue is not in the oral cavity
→ Thank you for your pointed out. We corrected Text “tongue” and “buccal mucosa”. And this study was not including base of tongue.
-Table 1 and 2 are redundant and I would keep Table 2 in the main text and move table 1 in supplementary
→ Thank you for your advice. As you pointed out, Table 1 was moved supplementary.
-Table 2: a relevant statistical test to evaluate the significance of the differences observed in the Control site and in the Lesion site needs to be added
→ As you pointed out, we added the a relevant statistical test.
-When an absolute number of patients is indicated throughout the text, a percentage should be added
→ We added a percentage throughout the text.
-ROC curves should be provided as supplementary material
→ We added ROC curves as supplementary.
-It is unclear how the coefficients in the “combination of each factors for detection of oral cancer” (lines 160-166, p11) were defined
→ We added L455-458 on Material and Methods.
-All the paragraph “2.4.1. Sub-analysis of oral cancer” should be removed or added to supplementary material, as it does not bring much and is not the main message of the paper
→ As you pointed out, paragraph 2.4.1 was delated.
-To many Figures are presented, they should be selected and the remaining moved to supplementary
→ Thank you for your advice, we selected Figure 1-5, and deleted Figure 6 and 7.
Round 2
Reviewer 1 Report
All the reviewers' concerns had been well responded.
Reviewer 2 Report
Please reference the clinicaltrial.gov trial by providing the NTC number in the Material and Methods section